# Development and Upscaling of SiO_2_@TiO_2_ Core-Shell Nanoparticles for Methylene Blue Removal

**DOI:** 10.3390/nano13162276

**Published:** 2023-08-08

**Authors:** Bárbara R. Gomes, Joana L. Lopes, Lorena Coelho, Mattia Ligonzo, Monica Rigoletto, Giuliana Magnacca, Francesca Deganello

**Affiliations:** 1CeNTItvc—Centre for Nanotechnology and Smart Materials, Vila Nova de Famalicão 4760-034, Portugal; brgomes@centi.pt (B.R.G.); jllopes@centi.pt (J.L.L.); 2Dipartimento di Chimica, Università degli Studi di Torino (UNITO), Via Pietro Giuria 7, 10124 Torino, Italy; mattia.ligonzo@edu.unito.it (M.L.); monica.rigoletto@unito.it (M.R.); giuliana.magnacca@unito.it (G.M.); 3NIS Interdepartmental Centre, Università degli Studi di Torino, Via Pietro Giuria 7, 10124 Torino, Italy; 4Consiglio Nazionale delle Ricerche (CNR) Istituto per lo Studio dei Materiali Nanostrutturati (ISMN), Via Ugo La Malfa 153, 90146 Palermo, Italy; francesca.deganello@cnr.it

**Keywords:** SiO_2_@TiO_2_, core-shell, upscaling, photocatalysis, dye removal, dye adsorption, water treatments

## Abstract

SiO_2_@TiO_2_ core-shell nanoparticles were successfully synthesized via a simple, reproducible, and low-cost method and tested for methylene blue adsorption and UV photodegradation, with a view to their application in wastewater treatment. The monodisperse SiO_2_ core was obtained by the classical Stöber method and then coated with a thin layer of TiO_2_, followed by calcination or hydrothermal treatments. The properties of SiO_2_@TiO_2_ core-shell NPs resulted from the synergy between the photocatalytic properties of TiO_2_ and the adsorptive properties of SiO_2_. The synthesized NPs were characterized using FT-IR spectroscopy, HR-TEM, FE–SEM, and EDS. Zeta potential, specific surface area, and porosity were also determined. The results show that the synthesized SiO_2_@TiO_2_ NPs that are hydrothermally treated have similar behaviors and properties regardless of the hydrothermal treatment type and synthesis scale and better performance compared to the SiO_2_@TiO_2_ calcined and TiO_2_ reference samples. The generation of reactive species was determined by EPR, and the photocatalytic activity was evaluated by the methylene blue (MB) removal in aqueous solution under UV light. Hydrothermally treated SiO_2_@TiO_2_ showed the highest adsorption capacity and photocatalytic removal of almost 100% of MB after 15 min in UV light, 55 and 89% higher compared to SiO_2_ and TiO_2_ reference samples, respectively, while the SiO_2_@TiO_2_ calcined sample showed 80%. It was also observed that the SiO_2_-containing samples showed a considerable adsorption capacity compared to the TiO_2_ reference sample, which improved the MB removal. These results demonstrate the efficient synergy effect between SiO_2_ and TiO_2_, which enhances both the adsorption and photocatalytic properties of the nanomaterial. A possible photocatalytic mechanism was also proposed. Also noteworthy is that the performance of the upscaled HT1 sample was similar to one of the lab-scale synthesized samples, demonstrating the potentiality of this synthesis methodology in producing candidate nanomaterials for the removal of contaminants from wastewater.

## 1. Introduction

A major challenge society will be facing during the twenty-first century is to supply and ensure safe water for the entire ecosystem. Rapid industrialization and population growth are the major causes of water pollution, introducing harmful organic pollutants into the environment, such as organic dyes, phenolic compounds, bacteria, viruses, and pharmaceutical products, among others [1].

To minimize the significant and serious impact of these wastewater pollutants on human life, it is crucial to develop sustainable technologies focused on their degradation through photocatalytic processes, due to the ability to generate reactive oxygen species. In this context, metal oxide nanomaterials appear as promising candidates [2].

Metal oxide nanomaterials (i.e., Fe_3_O_4_, TiO_2_, Al_2_O_3_, CuO, ZnO, and MgO) possess different physical, chemical, and morphological properties that can be tailored for a specific application, by strictly controlling the synthesis conditions and thus the surface-to-volume ratio, the particle size, and the defect [3]. In particular, TiO_2_ has been widely employed and extensively investigated due to its advantageous properties such as nontoxicity, high photocatalytic activity, low cost, and excellent oxidation resistance [4,5]. The photocatalytic properties of TiO_2_ result from the generation of charge upon exposure to UV rays with a wavelength corresponding to the bandgap of TiO_2_ [6,7]. Despite all these convenient features, the application of nanometric TiO_2_ is limited by the easy agglomeration of its NPs [8,9], the transformation of its crystalline phases [10,11], a reduction in the surface area after heat treatment [8,11,12], and the easy recombination of photogenerated electron–hole pairs [11]. Additionally, some risks are associated with the application of this nanomaterial due to the possible presence of residual nanoparticles in treated waters and their difficult separation, which involves additional purification steps, such as chemical and physical filtration, distillation, and reverse osmosis [13,14,15,16].

To maintain the photocatalytic properties of the powders and to prevent issues related to their release and recovery necessity, TiO_2_ and other functional metals oxides can be immobilized on the surface of a massive support, as beads, membranes, fibers, 3D porous structures, and organic or inorganic supports [17,18] can be deposited as a layer on the surface of ZrO_2_, SiO_2_, CeO_2_, Al_2_O_3_, CaO, Au, Ag, Cu, or Fe [7,19]. For example, silicon dioxide (SiO_2_) has been widely used due to its well-known surface chemistry, high thermal and mechanical stability, high adsorption capacity, low cost, and easy synthesis using the Stöber method [11]. It also exhibits high transparency in the UV/Vis region and is easily leached from core-shell structures with an alkaline solution [11]. It has been found that the addition of SiO_2_ shifts the polymorphic transformation of the anatase phase into rutile to higher temperatures and increases the surface area, resulting in a decrease in the particle size. Silica-titania photocatalytic nanocomposites have been also prepared in the form of nanopowder mixtures [20], nanofiber membranes [21], nanocoatings [22], colloidal nanocomposites [23], or in core-shell configurations [24].

The properties of the core-shell structures result from the integration of the unique properties of the two original materials. They can combine the properties of both the core and the shell, or they can show synergistic properties. Furthermore, particle stability and dispersion increase, and the shell material can provide easier surface modification and functionalization of the nanoparticles [24]. Additionally, they can be also employed to reduce the cost of an expensive material as only a small amount is needed to cover the shell [24]. An important example is the use of a titanium monolayer on a SiO_2_ surface that modifies its overall electronic structure, inducing a significant increase in photocatalytic activity [11]. In this sense, the system TiO_2_ anatase/SiO_2_ combines the superhydrophilic properties of SiO_2_ and the photocatalytic property of TiO_2_ anatase. The core@shell configuration corresponds to the combination of the enhanced properties of the core (SiO_2_) and of the shell (TiO_2_).

Despite the promising performance of TiO_2_-SiO_2_ nanomaterial combinations, the scalability of their fabrication procedure is still limited. One of the major barriers is the technology transfer to industry [25,26]. Regarding SiO_2_@TiO_2_ NPs, there are still poor solutions for their large-scale synthesis toward industrial production, and emerging strategies to improve the reproducible synthesis of large batches are needed to satisfy the future demand [26,27]. The hydrothermal method is regarded as a suitable synthetic approach and is one of the strategies adopted during the nanoparticle crystallization process. Typically, hydrothermal processes are extremely attractive for large-scale production, as they are environmentally friendly, allow the easy recovery of the photocatalyst after the synthesis, and do not require any post-calcination treatment, which allows access to photocatalytically active crystalline phases at much lower temperatures than those required for air calcination. Since the process occurs in an aqueous solution medium, the hydrophilicity of the material is improved due to the increase in surface hydroxyl groups [28].

In this paper, core (SiO_2_) and SiO_2_@TiO_2_ core@shell NPs were synthesized, characterized, and tested for methylene blue adsorption and UV photodegradation, evaluating the nanomaterials from the perspective of wastewater treatment applications, where the reactive species generation and dye removal via adsorption and photocatalytic processes are among the most important processes.

A comparison between calcination and hydrothermal methods was performed. One of the main objectives of the present work was to assess the viability of the synthesis scalability of the core@shell nanomaterial and its comparison with laboratory-scale synthesis. The synthesized SiO_2_@TiO_2_ core@shell NPs were characterized by complementary techniques such as powder X-ray diffraction (XRD), high-resolution transmission electron microscopy (HR-TEM), field emission scanning electron microscopy (FE–SEM), energy dispersive X-ray spectroscopy (EDS), dynamic light scattering (DLS), attenuated total reflectance Fourier transform infrared spectroscopy (ATR–FTIR), and N_2_ gas-volumetric adsorption–desorption at 77 K.

## 2. Materials and Methods

### 2.1. Reagents

Commercial reagents tetraethyl orthosilicate (TEOS) (98%, Sigma–Aldrich, Lisbon, Portugal), tetrabutyl orthotitanate (TBOT) (97%, Sigma–Aldrich, Portugal), ammonium hydroxide (30%, Labkem, Catalonia, Spain), and 2–isopropanol (99.8%, VRW Chemicals, Carnaxide, Portugal) were used as received. TiO_2_ P25 (>99.5%, 25 nm) was purchased from Evonik and ultrapure water (Milli-Q^®^, Darmstadt, Germany) and ethanol (EtOH) (99.8%, AGA, Prior Velho, Portugal) were used as solvents. Methylene blue powder (MB) was purchased from Sigma-Aldrich.

### 2.2. Synthesis of SiO_2_ and SiO_2_@TiO_2_ Core-Shell Nanoparticles

The synthesis process can be divided into the following steps: monodisperse SiO_2_ cores were synthesized via the classical Stöber method, and then the synthesized SiO_2_ cores were coated with a thin layer of TiO_2_, followed by calcination [28] or hydrothermal treatments [29,30]

For the synthesis of the SiO_2_ core, first, 64 mL of water, 320 mL of 2–isopropanol, and 16 mL of ammonium hydroxide were mixed. Then, 16 mL of TEOS was added dropwise with a peristaltic pump. The solution was kept at 60 °C under constant magnetic stirring. After reacting for 2 h, the solution was centrifuged (9000 rpm for 10 min) and washed twice with EtOH.

For the synthesis of SiO_2_@TiO_2_ core-shell nanoparticles, the synthesized SiO_2_ particles were dispersed in 200 mL of EtOH and in 2 mL of water, under continuous stirring until total dissolution. This mixture was kept in an ultrasonic bath at 40 °C for 15 min. Then, the mixture was heated to 85 °C and, when reached, 10 mL of TBOT was added dropwise with a peristaltic pump to 30 mL of EtOH and kept under reflux for 90 min. The particles were centrifuged (9000 rpm for 10 min) and washed with EtOH. After that, the particles were centrifuged and washed with water. The synthesized SiO_2_@TiO_2_ core-shell was dried at 60 °C for 1 h in the oven. The particles were subjected to different processes such as calcination or hydrothermal treatments for titanium crystallization to achieve the anatase and rutile phases.

The calcination treatment was carried out in an open crucible at 650 °C for 2 h with a heating ramp for 50 min (12 °C/min.). For the hydrothermal treatment, the synthesized SiO_2_@TiO_2_ NPs were dispersed in a 150 mL EtOH: water (molar ratio 1:1) solution (HT1) or in 150 mL of water (HT2). The process was carried out in a 200 mL Teflon-coated stainless-steel autoclave at 140 °C for 6 h, with a heating ramp for 30 min (5 °C/min.). The core-shell NPs obtained for HT2 were previously recovered by centrifugation (9000 rpm for 10 min) and washed twice with water. Appendix A shows a schematic of the main steps for the synthesis of core-shell NPs.

Additionally, a ten times upscale of the hydrothermal method 1 (SiO_2_@TiO_2_ HT1 upscaled) synthesis process was evaluated, following the procedure described previously. This represents a novelty concerning the applicability of this type of material for upscaling process applications like wastewater treatment.

All the samples were compared to pure SiO_2_ and TiO_2_ P25 during the characterization procedure.

### 2.3. Physical Chemical Characterization of SiO_2_ and SiO_2_@TiO_2_ Core-Shell Nanoparticles

Attenuated total reflectance Fourier Transform Infrared (FT-IR ATR) spectra were recorded in transmittance mode on a Perkin Elmer Spectrum 100 FT-IR ATR. Spectra were obtained with SPECTRUM software in the 4000–650 cm^−1^ range at a maximum resolution of 4 cm^−1^.

Zeta potential measurements were determined via DLS analysis with a Zetasizer Nano ZS90 (Malvern Panalytical) and a disposable capillary cell (DTS1070) at 25 °C. Zetasizer Software v7.13 was used for data acquisition. Nanoparticles were diluted in water to a concentration of 1 g/L, with pH values ranging from 4.9 to 6.9. All the measurements were performed in triplicate.

The morphology of the materials was examined using high-resolution transmission electron microscopy (HR-TEM) and field emission electron microscopy (FE–SEM).

HR-TEM micrographs were obtained using a JEOL JEM 3010 instrument (300 kV of acceleration potential) equipped with a LaB6 filament. For the specimen preparation, a few drops of powder water suspensions were supported on a 200-mesh carbon-coated copper grid and left to dry before analysis. The as-obtained images were analyzed with ImageJ software to measure interplanar distances of crystalline regions, particle sizes, and morphological features of the samples.

FE–SEM images were recorded by means of an FIB-FESEM/EBSD/TOF-SIMS Tescan S9000G microscope. The preliminary metallization of the samples was performed via the deposition of 5 nm of Cr using an Emitech K575X sputter coater equipped with a film thickness monitor.

The energy-dispersive X-ray spectroscopy (EDS) measurements were performed with AZtecLive and an ULTIM Max EDS System: DETECTOR OXFORD EDS Ultim Max—Software AZTEC.

The crystalline structures and phase identification of the synthesized NPs were evaluated using XRD analysis, carried out on a Bruker-Siemens D5000 X-ray powder diffractometer equipped with a Kristalloflex 760 X-ray generator and with a curved graphite monochromator and an X’Pert PRO MPD from PANalytical in Bragg−Brentano geometry diffractometers, both using Cu Kα radiation (40 kV/30 mA) and a flat sample-holder. The XRD pattern acquisition was performed in the 2θ range of 10−80° with 0.02° interval steps, 70 s step−1 to improve the signal-to-noise ratio.

The specific surface area (SSA) and porous properties of the materials were determined by N_2_ gas-volumetric adsorption at 77 K by means of ASAP2020 (Micromeritics). In prior analyses, all samples were outgassed in a vacuum (residual pressure < 10^−2^ mbar) to remove atmospheric contaminants adsorbed at the surface or inside the pores.

To determine the capacity of the materials for probe-molecule adsorption, the powders were crushed in an agate mortar, pressed in the form of self-supporting pellets (around 10 mg/cm^2^), protected in a holed gold frame, and put in a particular sample holder for pretreating the sample in a vacuum, contacting the materials with the probe molecules in the gas phase and recording the spectra. FT-IR spectra were obtained by using a Bruker Vector 22 spectrophotometer equipped with a Globar source and a DTGS detector. The spectra were recorded both in ATR (diamond cell) and transmission mode with 128 scans at a 4 cm^−1^ resolution in the 4000−400 cm^−1^ range.

### 2.4. Reactive Species Generation

The obtained nanomaterials were characterized for the generation of reactive species, one of the steps present during dye removal.

Electron paramagnetic resonance (EPR)-spin trapping measurements were performed to evaluate the hydroxyl radical generation to evaluate the potential of the NPs as an antibacterial material. A 210 ppm ultra-pure water suspension of a photocatalyst (SiO_2_@TiO_2_, SiO_2_@TiO_2_ HT1, SiO_2_@TiO_2_ HT2 or SiO_2_@TiO_2_ HT1 upscaled) was irradiated under simulated solar light (SOL2 honle UV technology, 380 nm filter) for 30 min, and then 5,5-dimethyl-1-pyrroline-N-oxide (DMPO) was added to reach a final concentration of 0.017 mM and left under irradiation for an additional 7 min. Finally, the samples were transferred in capillary quartz tubes, and the EPR spectra were recorded in an X-band Bruker EMX spectrometer using the following experimental parameters: microwave frequency 9.86 GHz, microwave power 5 mW, and modulation amplitude 1 Gauss.

### 2.5. Photocatalytic Removal of Dye

The adsorption capacity and photocatalytic properties of the synthesized nanomaterials were evaluated using methylene blue (MB) in aqueous solution as a model dye.

A stock solution of 10 mg/L MB was prepared by dissolving MB in water, and a particle catalyst suspension was prepared at 20 g/L by dissolving in MB aqueous solutions to be tested for the degradation of MB solution under UV light (VDL15UV 365 nm from EHQ POWER).

The samples were collected in appropriate time intervals (up to 8 h) and were filtered with a syringe filter (CHROMAFIL^®^ RC-45/25, regenerated cellulose, 0.45 µm) to remove the catalyst. In addition to visual analysis, the removal of MB was determined based on the absorption at 663 nm by using a UV/Vis spectrophotometer (Lambda35 from Perkin Elmer). The absorbance of samples was used to calculate the concentration using the calibration curve constructed based on the Beer–Lambert law. The removal efficiency was calculated from the following Equation (1) where C is the MB concentration for each sample (SiO_2_, TiO_2_, and SiO_2_@TiO_2_ NPs) for each analysis time and $C_{0}$ is the MB reference sample concentration for each time [31]. The adsorption capability ($q_{e})\;$was calculated following Equation (2), where V is the volume of MB solution in contact with the catalyst and W is the catalyst mass [32].
(1)$$\%Removal\;efficiency=(1-\frac{C}{C_{0}})\times 100$$
(2)$$q_{e}=\frac{\left(C_{0}-C\right)\times V}{W}$$

## 3. Results and Discussion

### 3.1. Physical Chemical Characterization of SiO_2_ and SiO_2_@TiO_2_ Core-Shell Nanoparticles

The prepared SiO_2_@TiO_2_ NPs were analyzed using FT-IR ATR. The spectra of the synthesized NPs are reported in Figure 1 together with those of the reference samples (pure SiO_2_ and TiO_2_ P25). According to the results obtained, the symmetric and asymmetric stretching modes of the Si-O-Si bond are visible around 790 and 1110 cm^−1^. These two peaks are observed for SiO_2_ and all SiO_2_@TiO_2_ samples, indicating that the TiO_2_ coating thickness is thin enough to detect the IR absorption signal from the core SiO_2_ [33,34,35]. However, it is also possible to observe the intensity of the Si-O-Si bond as being higher than calcined SiO_2_@TiO_2_ compared to the hydrothermal samples, which can be explained by the thickness of the TiO_2_ shell being lower for the calcined sample compared to the hydrothermal samples, according to the previous conclusion.

The peak present around 970 cm^−1^ is attributed to the Ti-O-Si bond structure, as it is present in all core-shell NPs and is not present in the reference samples. Additionally, the band around 1400 cm^−1^ was attributed to Ti-O-Ti vibration, observed for TiO_2_ and all SiO_2_@TiO_2_ samples, confirming the formation of the titanium shell bonded to the silica core [36,37]. Their intensity is higher for hydrothermal samples compared to calcined and reference samples, a feature that could be related to the thickness of the TiO_2_ shell [34].

Regarding the SiO_2_@TiO_2_ (HT1) upscaled sample, when compared with the one synthesized at the lab scale, it is possible to verify the same peaks relative to the bands identified above, showing that the NPs developed in the upscaled process present the expected composition when analyzed by this technique.

Table 1 shows the zeta potential of the synthesized SiO_2_ and SiO_2_@TiO_2_ core-shell NPs subjected to different treatments obtained by the DLS analyses.

The values indicated an overall negative surface charge of −20.5 ± 0.4 mV for the SiO_2_ particles. On the other hand, the calcined SiO_2_@TiO_2_ particles exhibit a small increase in the zeta potential of −21.4 ± 0.9 mV. The SiO_2_@TiO_2_ particles subjected to the different hydrothermal treatments show significant differences. Indeed, both SiO_2_@TiO_2_ HT1 and HT2 particles show a significant increase in zeta potential (−33.9 ± 0.4 mV and 37.3 ± 0.4 mV, respectively). These values are negative and high, indicating the high stability of the suspension in water and the low agglomeration of the SiO_2_@TiO_2_ HT1 and HT2 particles [38,39,40].

The difference between the zeta potential values for the SiO_2_ and all core-shell NP samples could be explained by the increase in hydroxyl groups on the surface, which also increases, negatively or positively, the zeta potential values, as reported in [41]. These results are also supported by the FT-IR analysis presented in Figure 1, where it is possible to observe one peak around 1600 cm^−1^ and another one at 3400 cm^−1^ representative of the O-H vibration band, which indicates the presence of the hydroxyl groups on the NPs’ surface. On the other hand, TiO_2_ is negatively charged [42], and a higher amount of size-controlled TiO_2_ NPs on the surface of the SiO_2_ core could be related to the differences observed [40].

In fact, it is possible to verify a higher negative zeta potential value for hydrothermal samples HT1 and HT2 compared to calcined, consistent with the increase in shell thickness, as represented by HR-TEM images.

Even though the SiO_2_@TiO_2_ HT1 upscaled sample has a value below that obtained at the lab scale, the result obtained is consistent with what was expected, in which the value is more negative when compared with the SiO_2_ core and SiO_2_@TiO_2_ calcined material.

Figure 2 shows the HR-TEM images. It is possible to observe that SiO_2_ particles (Figure 2a) show a spherical and smooth morphology with an average diameter of 184.9 ± 10.6 nm. The surface of the SiO_2_ particles is uniform, without any visible defects. However, the smooth surface of the SiO_2_ particles becomes slightly rough for the core-shell particles, suggesting that the formation of the TiO_2_ layer was successful (Figure 2b–d). It should be noted that the detected impurity surrounding the NPs could probably be due to some organic residues deriving from the synthesis or TEM preparation.

As can be observed in Figure 2b, the SiO_2_@TiO_2_ NPs prepared via the calcination method show a small increased average diameter (210.7 ± 33 nm) compared to SiO_2_ particles, which may be related to the crystalline TiO_2_ shell.

Figure 2c,d show the surface of the SiO_2_@TiO_2_ core-shell NPs after different hydrothermal treatments (HT1 and HT2, respectively), and the spherical surface of the SiO_2_ core is clearly visible, as well as the thin layer of the TiO_2_ shell around it. This result was also evidenced by the EDS map reported in Appendix A. For hydrothermal samples (HT1 and HT2), the average diameters of the SiO_2_@TiO_2_ for both samples are 332.5 ± 3.5 nm. The TiO_2_ shell has an average size of 35 nm. Appendix A shows the presence of a fringe pattern, within which a distance of 0.356 nm is ascribable to the anatase phase of TiO_2_ [43]. Furthermore, it is possible to verify that the shell formed in the hydrothermal treatment is larger than that formed by the calcination process, as evidenced by the images below.

The FE-SEM images below prove the results obtained by HR-TEM and described above. It is possible to observe the spherical well-formed particles for SiO_2_ (Figure 3a) with a smooth surface without any evident defects. In contrast, the synthesized SiO_2_@TiO_2_ core-shell NPs, both calcined (Figure 3b) and HT (Figure 3c,d), show a rough and textured surface, which suggests that the TiO_2_ was successfully coated on the silica particles, as also confirmed by the presence of Si and Ti elements evidenced by the EDS analyses (Table 2). Most of the detected impurity is made of C that can be associated with unreacted precursors but also with the adhesive conductive carbon tape used for fixing the materials to the sample holder. It is worth evidencing that the upscaled sample presents a higher amount of Ti with respect to the Si amount (Ti 11.2 vs. Si 8.1), suggesting a more efficient coverage of the SiO_2_ cores.

Figure 3e,f show the obtained images for SiO_2_@TiO_2_ core-shell NPs (HT1) upscaled. Analyzing the images, it is possible to verify that the morphology of the obtained NPs is similar to that obtained at the lab scale (Figure 3c). These results suggest that the upscaling process was successful, proving it is feasible and reproducible, as these aspects are known to be the main identified challenges in nanoparticle synthesis development. It is worth evidencing that the upscaled sample presents a very high amount of Ti with respect to the Si amount (Ti 11.2 vs. Si 8.1), suggesting a more efficient coverage of the SiO_2_ cores.

The synthesized SiO_2_ and SiO_2_@TiO_2_ NPs show broad XRD peaks typical of amorphous and/or nanometric powders [44,45], while commercial TiO_2_ is a fully crystalline powder composed of the anatase and rutile phases [46]. XRD patterns of commercial titania show diffraction peaks attributed to the anatase (2θ ≈ 25°), rutile (2θ ≈ 28°), and mixed phases of TiO_2_, as reported by El-Desoky et al. [47]. In Figure 4, both the SiO_2_ (red pattern) and TiO_2_ (light blue pattern) phases are clearly recognizable in the green pattern of SiO_2_@TiO_2_. In particular, the XRD pattern of SiO_2_@TiO_2_ shows an evident diffraction peak (2θ ≈ 25°), which corresponds to the anatase phase. The very different microstructure and morphology of the TiO_2_ in the shell compared to the commercial TiO_2_ powder indicates that titania in the core-shell material is highly nanocrystalline.

The mixed anatase–rutile phases are also evident in the 2θ = 60–80° range of hydrothermally treated SiO_2_@TiO_2_ core-shell materials (HT1 and HT2). The XRD pattern of the SiO_2_@TiO_2_ HT1 upscaled sample indicates diffraction peaks at 2θ ≈ 25° and 2θ ≈ 47.5° corresponding to the anatase phase, just like the correspondent lab-scale HT1 sample, suggesting that the upscaling process was successful.

The main miller indexes (101), (200), and (200), corresponding to the anatase and rutile phases of TiO_2_ (JCPDS 75-1537 reference), respectively, are indicated by vertical dashed lines in the XRD results in Figure 4.

N_2_ gas-volumetric 77 K adsorption/desorption analyses were performed for SiO_2_, TiO_2_, and SiO_2_@TiO_2_ core-shell NPs to evaluate their textural properties (specific surface area (SSA) and porosity). The main results are reported in Table 3 and Figure 5 and Figure 6. SiO_2_ shows an SSA of 18 m^2^/g. In contrast, the TiO_2_ and the calcined SiO_2_@TiO_2_ particles show an SSA with higher values (52 m^2^/g in both cases). The increase in SSA suggests the presence of nanocrystalline TiO_2_ particles, since they have a larger surface area due to their small size, as already evidenced by TEM images reported in Figure 2. In addition, the presence of small aggregated TiO_2_ nanoparticles causes the formation of pores, essentially in the range of large meso- and macropores, as depicted in Figure 5b.

The synthesized SiO_2_@TiO_2_ NPs (HT1 and HT2) show isotherms and pore size distribution curves reported in Figure 6**.** HT1 and HT2 show an SSA of 304 and 394 m^2^/g, respectively. The significant increase in SSA may be directly associated with the type of treatment (hydrothermal treatment vs. calcination), thus it seems the hydrothermal treatment contributes to the formation of smaller TiO_2_ particles in the shell of the composite materials with the consequent increase in SSA. The nitrogen adsorption–desorption isotherms are of the IV type [28]. According to the DFT analysis of the pore size, and with respect to the SiO_2_@TiO_2_ sample, it is possible to confirm an increase in the micropores and meso/macropores, as indicated by the results reported in Table 3 and Figure 6. According to the DFT analysis of the pore size, and with respect to the SiO_2_@TiO_2_ sample, it is possible to confirm an increase in the micropores and meso/macropores, as indicated by the results reported in Table 3 and Figure 6.

Figure 7a reports the transmittance spectra of SiO_2_@TiO_2_ in the form of self-supporting pellets in the presence of air and outgassed from RT up to 400 °C. The spectra can be described by dividing them into two regions [37,48]: (a) in the range between 4000 and 2500 cm^−1^, the surface OH groups, interacting via hydrogen bonding with each other and with adsorbed water molecules, are responsible for the huge absorption observed. Additionally, in the same range, CH groups of unreacted organic precursors or deriving from atmospheric contamination vibrate at around 3000 cm^−1^; (b) below 2100 cm^−1^, the vibrational signals of Si-O-Si of the bulk (about 800 and 1100 cm^−1^) and the correlated harmonic and combination modes (about 1600, 1900, and 2000 cm^−1^) absorb the main part of the radiation. In this range, signals of adsorbed water molecules at 1630 cm^−1^ (δHOH) and carbonate-like groups derived from the interaction of atmospheric CO_2_ with basic O^2−^ sites at 1900 and 1400 cm^−1^ appear. Another prominent but not useful signal is present at 2345 cm^−1^ due to the roto vibrational profile of some gaseous CO_2_ present in the spectrophotometer sample chamber.

During the outgassing process, physisorbed and chemisorbed molecules derived from the interaction of the material with gaseous molecules present in the atmosphere (mainly water and carbon dioxide) desorb from the surface, increasing the transparency of the sample (arrow in Figure 7a). In the high-frequency region, the decrease in hydrogen bonding interactions makes visible the signals of free SiOH groups at 3750 cm^−1^. The presence of this absorption suggests TiO_2_ NPs do not completely cover the surface of SiO_2_, and this could be beneficial for the photocatalytic activity of TiO_2_ because silanols can promote oxygen adsorption [49]. Moreover, the dehydration of the sample continues up to 400 °C of vacuum outgassing temperature, as witnessed by the change in the optical transparency of the material, and this indicates that the material is extremely hydrophilic, as expected for the presence of highly dispersed TiO_2_ NPs and for the presence of the polar interface where Si-O-Ti groups are formed. This feature surely enhances the capacity of the composite materials in interacting with polar substrates, i.e., methylene blue molecules.

To better evidence the behaviors of the SiO_2_/TiO_2_ interface, NH_3_ was used as a probe to investigate the material Lewis acidity (due to coordinatively unsaturated Ti^4+^ surface sites) and Brønsted acidity (due to Si-OH-Ti groups, i.e., OH groups present at the interface of SiO_2_/TiO_2_) [50]. The sample chosen for this investigation, SiO_2_@TiO_2_, was preliminarily outgassed at 120, 250, and 400 °C to almost completely remove the adsorbed atmospheric contaminants and create a surface that is reactive towards the interaction with the probe. Only the results obtained for the sample outgassed at 400 °C are reported for the sake of brevity, as they are not so different from the other results. The main spectra, registered in transmittance after the admission of the increasing pressure of NH_3_ in the cell, are reported in Figure 7b.

The presence of NH_3_ causes an important decrease in the optical transparency (arrow in Figure 7b) of the sample with the formation of signals related to NH^4+^ species (1450 cm^−1^) and NH_3_ interacting with Lewis acidic sites of material (1630 cm^−1^), both confirmed by the presence of the signal at 3300 cm^−1^. Moreover, the isosbestic point at 3520 cm^−1^ indicates that free OH groups progressively interact with increasing doses of NH_3_ molecules forming H-bonds. All these interactions depend on the NH_3_ pressure; therefore, a decrease in the amount of NH_3_ in the cell causes a partial recovery of the original profile of the spectrum, indicating a partially irreversible interaction of the molecule with the surface of the sample.

### 3.2. Reactive Species Generation

EPR spectra after irradiation under simulated solar light were recorded for all the samples in the presence of the spin-trap DMPO. In all cases (Figure 8) the typical spectral pattern of the DMPO-OH adduct with a_N_ = a_H_ ≅ 15.1 Gauss was observed, confirming the generation of strongly oxidizing hydroxyl radicals after the irradiation already observed in TiO_2_/SiO_2_ composite coatings and in many other hybrid materials containing TiO_2_ particles [51,52]. EPR experiments showed the production of a similar number of radical species for all the SiO_2_@TiO_2_ samples, indicating that the generation of hydroxyl radicals under irradiation is not dependent on the different preparation methodologies used in this work. This result shows that the hydrothermal treatment can produce a crystalline material under more ecofriendly conditions and is able to generate similar types and concentrations of reactive species compared to the nanomaterial obtained via calcination. Additionally, the upscaled material also presented results that confirm the feasibility of the scale-up process, since it was able to generate the same type and concentration of reactive species as the material synthesized at the laboratory scale.

### 3.3. Photocatalytic Removal of Dye

The photocatalytic removal efficiency of the prepared NPs was examined through the ability of the particles to remove MB under UV light irradiation. In this photocatalytic study, the pure SiO_2_, TiO_2_ nanoparticles, and an MB solution were used as references for comparison. After the UV light exposure, the samples were filtered and the total concentration of the MB was determined from the maximum absorption measurements using UV/Vis spectroscopy, as the characteristic peak of the MB dye at 663 nm is typically used to study its catalytic degradation [53].

Typically, when the dye solution is added into the mixture, the dye molecules start to adsorb on the surface of the solid catalyst particles, which decreases at a certain level of MB concentration [54].

The photocatalytic process through TiO_2_ typically involves the electrons in the conduction band, which participate in the reduction process and induce the reaction of molecular oxygen present in the atmosphere with the superoxide radical anion formation. The hydroxyl radicals, generated between the TiO_2_ surface and the adsorbed water molecules, and the superoxide ions, highly reactive, degrade the organic compounds through oxidative reactions [7,36,55]. In the case of MB, the degradation occurs through the oxidative process upon UV light irradiation [7]. Several authors have reported the development of different nanomaterials for the degradation of MB in aqueous solution [56,57,58,59].

As shown in Table 4, the MB reference sample did not exhibit significant differences in MB tonality and degradation over exposure time to UV radiation.

Visually comparing the samples prior to filtration, in the case of core-shell NPs, the samples prepared with hydrothermal treatment showed an initial difference in MB tone (0 min) when compared to the calcined sample, exhibiting a slightly dark blue-purple color, becoming more pronounced after exposure to UV light.

To perform a quantitative assessment of the removal efficiency, the suspensions were filtered to remove the catalyst; then, the supernatants were analyzed via UV/Vis spectroscopy to determine the MB removal. The obtained values were used to determine the MB concentration through a standard calibration curve (Equation (3)).
(3)y=0.145x+0.015

The different samples were compared to the MB reference sample with the respective time of exposure to UV radiation. It must be highlighted that the filtration of methylene blue did not decrease its concentration in solution; thus, the decrease in the dye concentration (i.e., the dye removal) was the result of the adsorbent and/or photocatalytic effects of the materials and not the effect of dye filtration. Figure 9a shows the MB removal efficiency, the MB mass removed from the solution during the exposure time, and Appendix A shows the images of the different sample solutions, for the same representative times, after UV light exposure and filtration.

For bare SiO_2_, the calculated removal efficiency demonstrates a constant MB decrease during the first 60 min with an efficiency removal of 45%. After 180 min of exposure, a significant increase in MB removal is observed and stabilizes over time, with a removal efficiency of between 69 and 74% after 480 min. These values can be explained by the adsorption capacity of this material type. Indeed, the capacity to remove organic dyes utilizing adsorbents and adsorption through physical methods has been investigated recently and this is a reported characteristic of SiO_2_ particles due to their porosity [60].

It is already known that to increase the organic dyes’ removal, the surface of porous materials like SiO_2_ can be modified with metal or metal oxide materials to improve the adsorption capacity and add photocatalytic activity to increase the degradation of organic compounds [60]. This behavior can be observed on the SiO_2_@TiO_2_ core-shell NPs’ results. After 15 min of exposure, the MB removal capacity is 80% in the case of the SiO_2_@TiO_2_ prepared using the calcination method, and almost complete for the samples prepared using hydrothermal methods, indicating a synergetic effect between the adsorption at the SiO_2_ core and the photocatalytic MB degradation by the TiO_2_ shell. Porous structured materials with a high surface area and photocatalytic capacity have been often commonly chosen for wastewater purification [7,60].

According to the obtained results, the possible MB degradation mechanism for these materials involves, in the first step, dye adsorption on the catalyst surface and its respective photodegradation after exposure to UV radiation [54,61]. Although a considerable removal of the dye is observed in the dark conditions, it should be noted that this only becomes possible after a separation/filtration process, as can be seen in the images in Table 4 and Appendix A. Before UV exposure (0 min) and filtration, all the solutions show turbidity and a more pronounced MB tonality, which decreases after the filtration step. Additionally, after the 8 h UV exposure, the samples become clearer and less turbid, indicating the photodegradation of the dye. It should also be noted that after filtration, only the hydrothermally treated samples show an almost complete removal of MB at 0 min, compared to the sample treated by calcination and the TiO_2_ reference sample. This may be due to the thickest TiO_2_ shell of the HT1 and HT2 samples compared to the calcined sample [62].

Based on the results represented in Figure 9b, it is possible to observe that the degradation rate of the SiO_2_@TiO_2_ NPs with hydrothermal treatment is considerably higher compared to the SiO_2_ and TiO_2_ reference samples (55 and 82%, respectively). This means that a considerable part of the MB was degraded at the initial exposure with only 15 min; this rate decreases over time, demonstrating that prolonged exposure produces the same results. The behavior of the TiO_2_ reference sample should be highlighted, in which the degradation rate is lower than all the samples under study, even the reference SiO_2_ sample, presenting the same degradation rate after 30 min. These results prove once again the synergistic and efficient adsorbent effect of SiO_2_ on MB degradation.

It is also to be noted that even in the dark, high percentages of MB removal are observed, being higher for SiO_2_@TiO_2_ samples prepared via the hydrothermal treatment, followed by the SiO_2_@TiO_2_ sample prepared using the calcination method, and then the bare SiO_2_. The same behavior was observed in the study of Urbashi Mahanta et al. (2022) where SiO_2_-TiO_2_ nanoparticles with a mole ratio of 5:1 showed the highest adsorption capacity of 88.6% after 30 min under dark conditions [7]. It is described in the literature that the combination of SiO_2_ and TiO_2_ is to enhance the photocatalytic performance of TiO_2_ by reducing the particle size of TiO_2_, improving the surface area, and increasing the thermal stability [7]. It was found that the prime factor enhancing the efficiency of the photocatalyst is its surface properties such as surface charge, porosity, and surface area [7].

For bare SiO_2_, the calculated removal efficiency demonstrates a constant MB decrease during the first 60 min with an efficiency removal of 45%. After 180 min of exposure, a significant increase in MB removal was observed and stabilizing over time, with a removal efficiency of between 69 and 74% after 480 min. These values can be explained by the adsorption capacity of this material type. Indeed, the capacity to remove organic dyes utilizing adsorbents and adsorption through physical methods has been investigated recently and this is a reported characteristic of SiO_2_ particles due to their porosity [60].

It is already known that to increase the organic dyes’ removal, the surface of porous materials like SiO_2_ can be modified with metal or metal oxide materials to improve the adsorption capacity and add photocatalytic activity to increase the degradation of organic compounds [60]. This behavior can be observed on the SiO_2_@TiO_2_ core-shell NP results. After 15 min of exposure, the MB removal capacity is 80% in the case of the SiO_2_@TiO_2_ prepared by the calcination method, and almost complete for the samples prepared by the hydrothermal methods, indicating a synergetic effect between the adsorption at the SiO_2_ core and the photocatalytic MB degradation by the TiO_2_ shell. Porous structured materials with a high surface area and photocatalytic capacity have been often commonly chosen for wastewater purification [7,60].

According to the obtained results, the possible MB degradation mechanism for these materials involves, in the first step, dye adsorption on the catalyst surface and its respective photodegradation after exposure to UV radiation [54,61]. Although a considerable removal of the dye is observed in dark conditions, it should be noted that this only becomes possible after a separation/filtration process. As can be seen in the images in Table 4, before UV exposure (0 min), all the solutions show turbidity and a more pronounced MB tonality. After 8 h of UV exposure, the samples become clearer and less turbid, indicating the photodegradation of the dye.

It is well known that the specific surface area has an important role in increasing photocatalytic activity [7]. The high surface area provides a number of active centers that can adsorb a large number of pollutant molecules [63]. According to the SSA results (Table 3), the surface area of the SiO_2_@TiO_2_ HT1 and HT2 samples are 94–95% higher than SiO_2_ and 83–87% higher than calcined SiO_2_@TiO_2_ and the TiO_2_ reference sample. These results confirm the effect of SiO_2_ on the improvement of specific surface area that directly impacts the MB degradation and removal. The high photocatalytic activities and high adsorption ability for organic contaminants demonstrate that the nanocomposite of SiO_2_-TiO_2_ is a promising candidate material for the future treatment of contaminated water to remove the contaminants effectively even without illumination.

Table 5 shows the initial adsorption capability of the samples in dark conditions. It is possible to observe once again the highest adsorption for the samples obtained via the hydrothermal method, followed by the calcined method and SiO_2_ reference sample. The TiO_2_ is the sample with the lowest MB adsorption performance, which agrees with the results demonstrated so far.

On the other hand, the TiO_2_ P25 reference material did not exhibit the same efficiency, starting with 11% in the dark, showing only an 18% removal efficiency after 15 min, stabilizing around 28–34% until 180 min, and increasing to 82% after 480 min. Typically, TiO_2_ reported in the literature presents lower MB degradation efficiency [36]. However, a comparison of the results with the literature is difficult. Even if the same catalyst is used, the parameters of the photocatalytic testing can be very different [64]. The values of the maximum degradation and removal efficiency of MB are listed in Table 6 with some experimental information to be compared with the obtained results of the present work.

Although TiO_2_ is the most used semiconductor material for photocatalysis due to its chemical stability [64], this work demonstrates that a SiO_2_@TiO_2_ core-shell structure improves the dye degradation by taking leverage of the adsorption capacity of SiO_2_.

It should be highlighted that in the present study, the photocatalytic degradation of MB occurred at a much lower power (15 W), in contrast to those found in most literature studies which used higher UV light power, as the MB removal is potentiated by the hydrothermally treated NPs.

In addition, once again, the viability of the SiO_2_@TiO_2_ (HT1) upscaled material synthesis and use was confirmed by the obtained results, where the removal and degradation of the MB behavior were the same when compared with the laboratorial scale material.

## 4. Conclusions

SiO_2_@TiO_2_ core-shell nanoparticles were obtained through the synthesis of a monodispersed SiO_2_ synthesized by the classical Stöber method, which was coated with a thin layer of TiO_2_, followed by calcination or hydrothermal treatment. The nanoparticles were tested for methylene blue removal and photocatalytic degradation under low-power UV light. The low temperature (140 °C) of the hydrothermal treatments (HT1 and HT2) was sufficient for the transformation of the titania amorphous phase into the anatase phase, allowing good crystallinity of the shell. The high negative value obtained for the zeta potential indicates that both the calcined SiO_2_@TiO_2_ and the hydrothermally treated SiO_2_@TiO_2_ (HT1 and HT2) are highly stable in water and exhibit low agglomeration.

EPR experiments showed that the generation of hydroxyl radicals under sunlight irradiation is not dependent on the type of treatment (calcination or hydrothermal) used for the TiO_2_ crystallization step. These results demonstrated that nanomaterials prepared via a hydrothermal treatment, a more ecofriendly condition, show the same performance as nanomaterials synthesized via the calcination method or even the commercial titania used as a reference.

The NPs prepared using hydrothermal methods showed the highest MB degradation capacity of almost 100% after 15 min when compared to the SiO_2_@TiO_2_ and TiO_2_ with 80 and 18%, respectively. Even under dark conditions, high percentages of MB removal were observed. This may be due to the thickest TiO_2_ shell of the HT1 and HT2 samples compared to the calcined sample and the higher SSA obtained via the hydrothermal method associated with the adsorption capacity of SiO_2_, which positively influences the photocatalysis capacity. The SiO_2_@TiO_2_ HT1, laboratory and upscaled samples, and HT2 presented the highest adsorption capacity, followed by SiO_2_@TiO_2_ calcined samples, the SiO_2_ reference sample, and, finally, the TiO_2_ reference sample.

The standout performance of the upscaled sample (SiO_2_@TiO_2_ (HT1) upscaled) indicates the viability of this solution for large-scale applications, where larger amounts of materials are needed.

The results obtained in this work have shown the potential of the SiO_2_@TiO_2_ core-shell particles as putative candidates for the removal of organic dyes from wastewaters, even without illumination.

## Figures and Tables

**Figure 1 nanomaterials-13-02276-f001:**
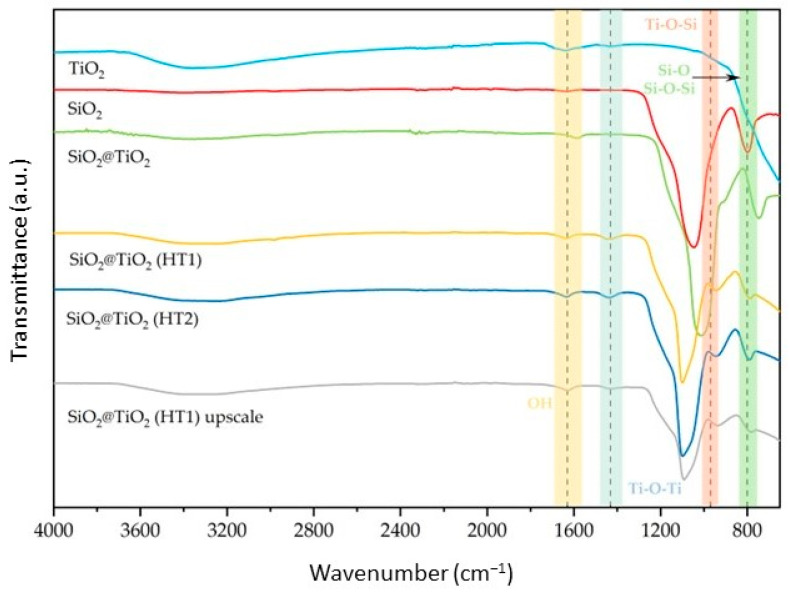
Normalized FT–IR ATR spectra of the synthesized SiO_2_@TiO_2_ NPs compared with pure SiO_2_ and TiO_2_.

**Figure 2 nanomaterials-13-02276-f002:**
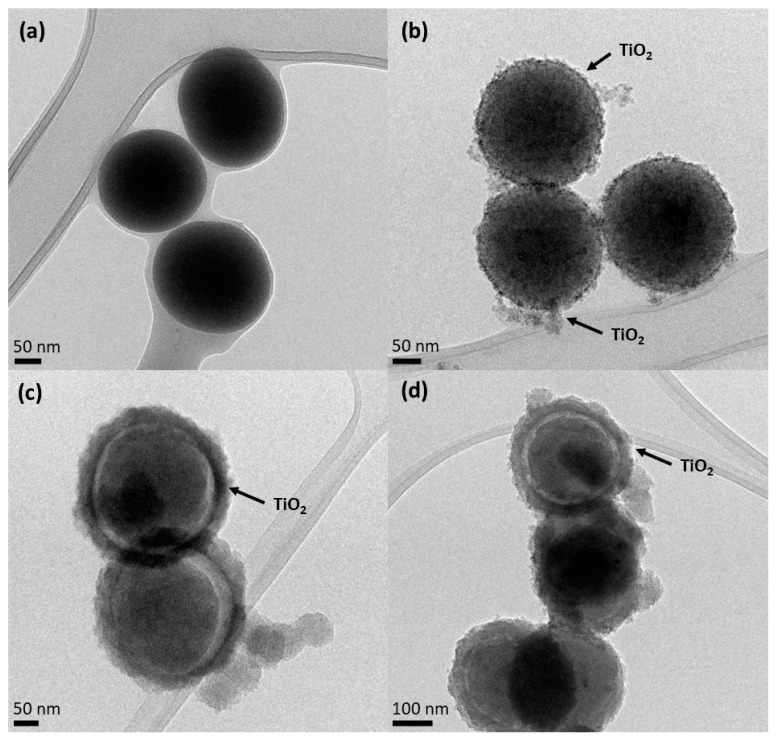
HR-TEM images of (**a**) SiO_2_ core, (**b**) SiO_2_@TiO_2_ calcined core-shell NPs, (**c**) SiO_2_@TiO_2_ core-shell NPs HT1, and (**d**) SiO_2_@TiO_2_ core-shell NPs HT2.

**Figure 3 nanomaterials-13-02276-f003:**
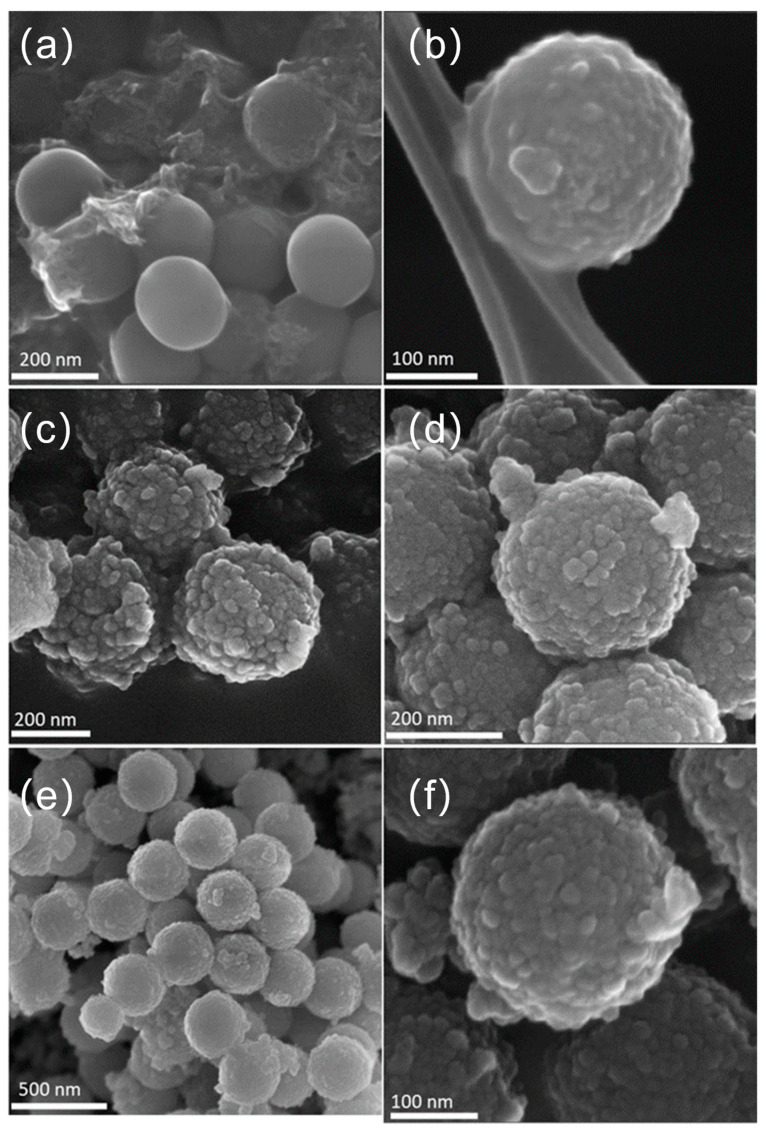
FE-SEM images of (**a**) SiO_2_ particles, (**b**) calcined SiO_2_@TiO_2_ core-shell NPs, (**c**) SiO_2_@TiO_2_ (HT1), (**d**) SiO_2_@TiO_2_ (HT2), and (**e**,**f**) SiO_2_@TiO_2_ (HT1) upscaled.

**Figure 4 nanomaterials-13-02276-f004:**
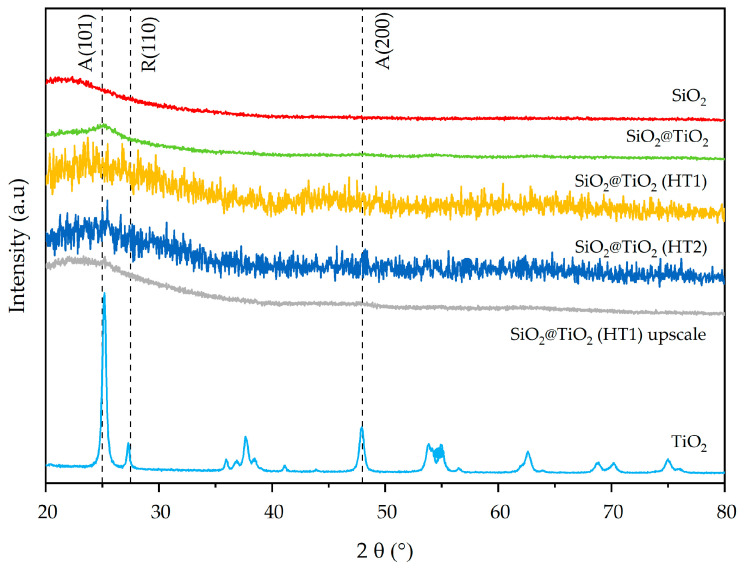
X-ray diffraction patterns of synthesized SiO_2_@TiO_2_ NPs compared with pure SiO_2_ and TiO_2_.

**Figure 5 nanomaterials-13-02276-f005:**
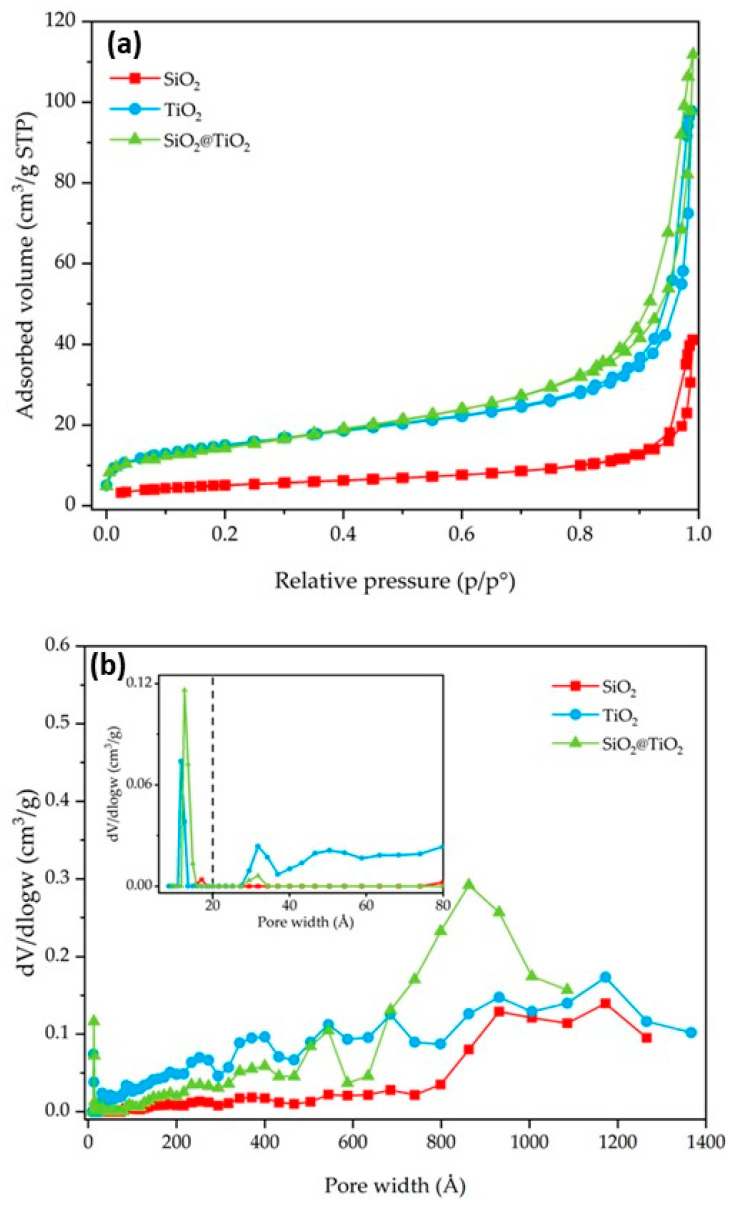
SiO_2_, TiO_2_, and SiO_2_@TiO_2_ particle results of (**a**) nitrogen adsorption–desorption isotherms and (**b**) pore size distribution. The broken vertical line in (**b**) represents the threshold value between micro- and meso/macropores.

**Figure 6 nanomaterials-13-02276-f006:**
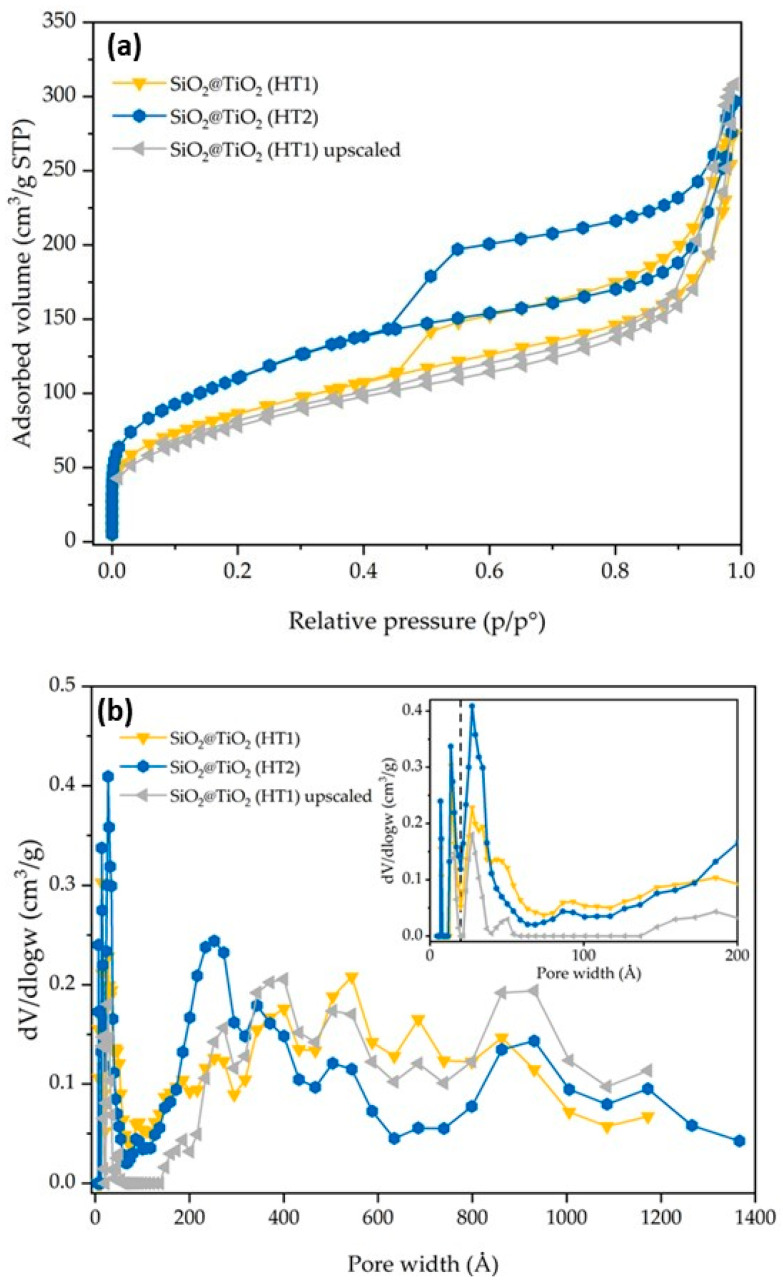
SiO_2_@TiO_2_ (HT1), SiO_2_@TiO_2_ (HT2), and SiO_2_@TiO_2_ (HT1) upscaled particles results of (**a**) nitrogen adsorption–desorption isotherms and (**b**) pore size distribution. The broken vertical line in (**b**) represents the threshold value between micro- and meso/macropores.

**Figure 7 nanomaterials-13-02276-f007:**
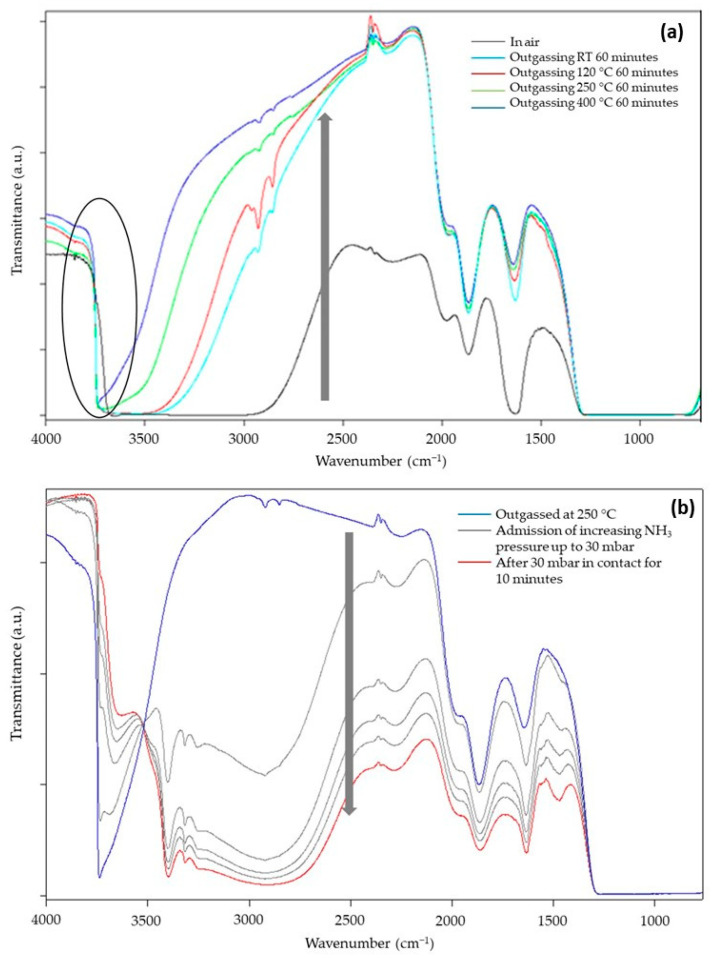
FT-IR transmittance spectra of SiO_2_@TiO_2_ (**a**) in contact with air and after outgassing in vacuo for 1 h at RT, 120, 250, and 400 °C and (**b**) outgassed at 250 °C, after the admission of increasing pressure of NH_3_.

**Figure 8 nanomaterials-13-02276-f008:**
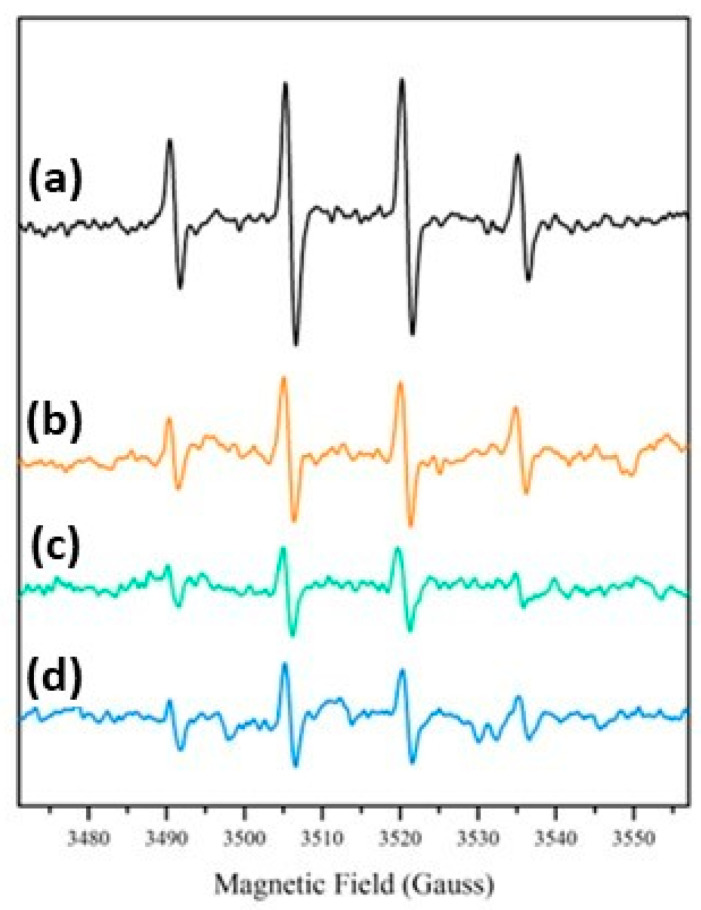
DMPO-OH EPR spectra after irradiation under simulated solar light: (a) SiO_2_@TiO_2_; (b) SiO_2_@TiO_2_ (HT1); (c) SiO_2_@TiO_2_ (HT2); and (d) SiO_2_@TiO_2_ (HT1) upscaled.

**Figure 9 nanomaterials-13-02276-f009:**
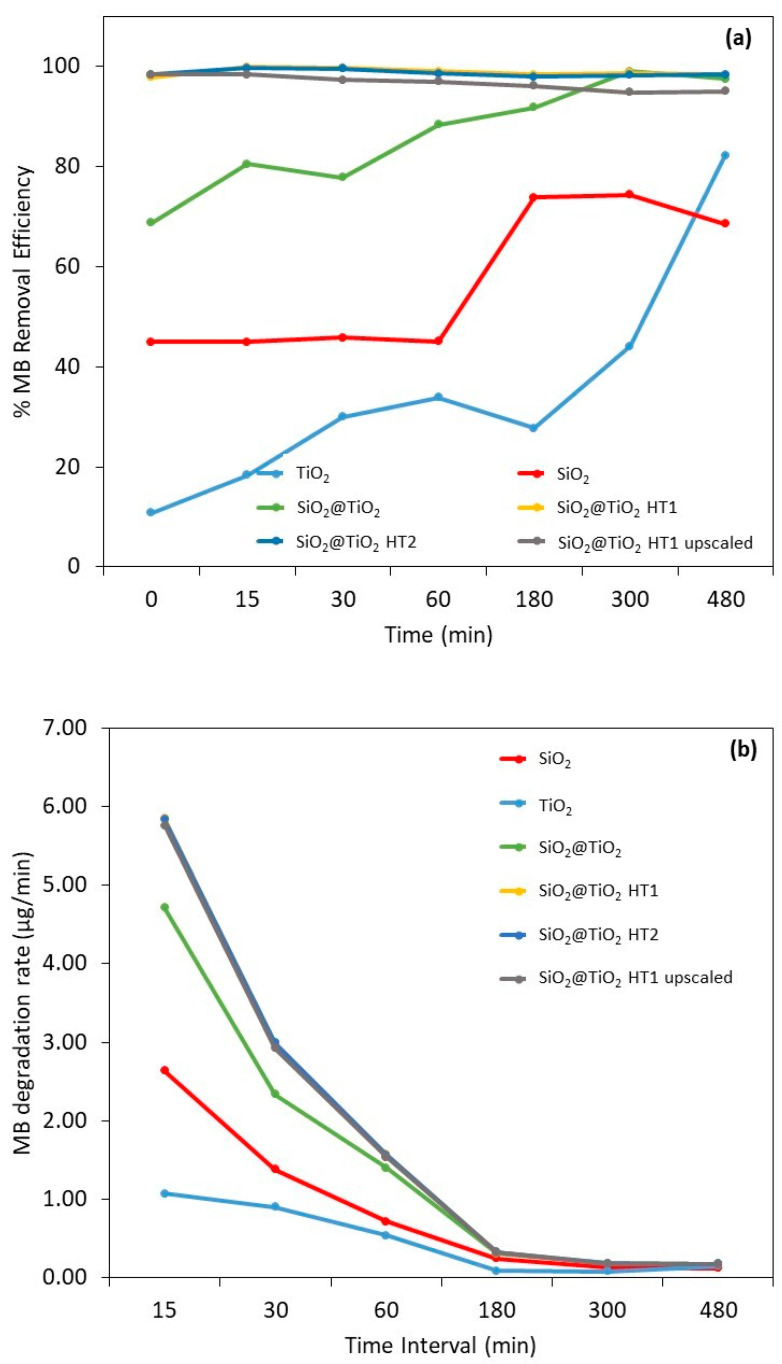
Results of (**a**) methylene blue removal efficiency and (**b**) degradation rate (µg/min) after 0, 15, 30, 60, 180, 300, and 480 min after UV radiation exposure. The vertical broken line represents the beginning of irradiation. The yellow experimental points of SiO_2_@TiO_2_ (HT1) are hidden by the experimental points of SiO_2_@TiO_2_ (HT2).

**Table 1 nanomaterials-13-02276-t001:** Zeta potential for the synthesized NPs.

Samples	Zeta Potential (mV)	pH	T (°C)
SiO_2_	−20.5 ± 0.4	5.6	25
SiO_2_@TiO_2_	−21.4 ± 0.9	4.9	
SiO_2_@TiO_2_ (HT1)	−33.9 ± 0.4	6.9	
SiO_2_@TiO_2_ (HT2)	−37.3 ± 0.5	6.6	
SiO_2_@TiO_2_ (HT1) upscaled	−28.1 ± 0.7	5.9	

**Table 2 nanomaterials-13-02276-t002:** EDS analysis median elements in each sample.

Samples	Si %wt	Ti %wt	O %wt	C+ Impurities
SiO_2_	9.9 ± 0.1	-	68.0 ± 0.2	21.2 ± 0.2
SiO_2_@TiO_2_	0.7 ± 0.0	0.5 ± 0.1	72.1 ± 0.5	26.7 ± 0.3
SiO_2_@TiO_2_ (HT1)	3.9 ± 0.1	1.9 ± 0.1	26.1 ± 0.4	67.5 ± 0.4
SiO_2_@TiO_2_ (HT2)	5.0 ± 0.1	0.9 ± 0.0	34.9 ± 0.4	59.1 ± 0.4
SiO_2_@TiO_2_ (HT1) upscaled	8.1 ± 0.2	11.2 ± 2.4	29.5 ± 0.8	51.0 ± 1.4

**Table 3 nanomaterials-13-02276-t003:** Textural parameters of the synthesized NPs obtained via BET analyses.

Samples	BET SSA (m^2^/g)	V Micropore (cm^3^/g)	V Meso/Macropore (cm^3^/g)	V Total (cm^3^/g)
SiO_2_	18	-	0.04	0.04
TiO_2_ P25	52	0.01	0.10	0.11
SiO_2_@TiO_2_	52	0.01	0.07	0.08
SiO_2_@TiO_2_ (HT1)	304	0.05	0.20	0.25
SiO_2_@TiO_2_ (HT2)	394	0.07	0.23	0.30
SiO_2_@TiO_2_ (HT1) upscaled	280	0.04	0.14	0.18

**Table 4 nanomaterials-13-02276-t004:** Images of the vials containing the powdered sample and the MB solution prior to filtration, before (0 min), and after 480 min of exposure to UV radiation.

	SiO_2_@TiO_2_
	SiO_2_ TiO_2_ Calcined HT1 HT2 HT1 US ^1^ MB
0 min	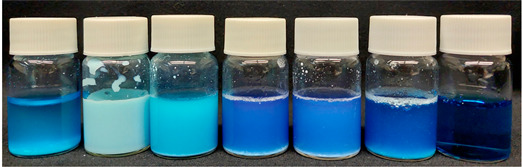
480 min	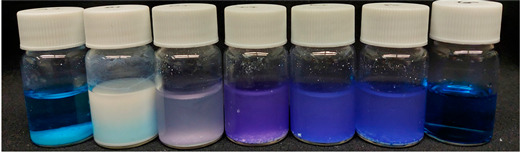

^1^ US—upscaled.

**Table 5 nanomaterials-13-02276-t005:** Adsorption capability of the samples in dark conditions.

Samples	Adsorption (µg/g)
SiO_2_	328
TiO_2_	79
SiO_2_@TiO_2_	502
SiO_2_@TiO_2_ (HT1)	714
SiO_2_@TiO_2_ (HT2)	718
SiO_2_@TiO_2_ (HT1) upscaled	718

**Table 6 nanomaterials-13-02276-t006:** Comparison of MB degradation with SiO_2_-TiO_2_ samples under different conditions.

#	Material	Exposure Time (min)	Conditions	Efficiency (%)	Ref.
1	SiO_2_-TiO_2_ nanoparticles	30	Dark	88.6	[7]
			Sunlight	90	
			UV light	85	
		120	Sunlight	98	
2	SiO_2_-TiO_2_ composite	360	UV light	90	[65]
3	TiO_2_–SiO_2_ hollow nanospheres	120	UV light	>90	
4	TiO_2_ prior calcination	300	UV light	7	[36]
	TiO_2_ after calcination			30	
	SiO_2_-TiO_2_ prior calcination			43	
	SiO_2_-TiO_2_ after calcination			76	
5	SiO_2_	30	UV light	45.8	This work
	TiO_2_			30.0	
	SiO_2_@TiO_2_ calcined			77.8	
	SiO_2_@TiO_2_ HT1 upscaled			97.2	

## Data Availability

The data presented in this study are available on request from the corresponding author. The data are not publicly available due to the intellectual property protection of the developments since the project is ongoing.

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
