# Peer review of "Development and Upscaling of SiO2@TiO2 Core-Shell Nanoparticles for Methylene Blue Removal"

_nanomaterials, 2023, doi:10.3390/nano13162276_

Round 1
Reviewer 1 Report
The present manuscript reported the preparation of SiO2@TiO2 core-shell nanocomposite and its application as efficient adsorbent for photocatalytic dye removal. The material was well characterized, and the experiments were designed well. I can recommend it for publication after some revisions.
1. The novelty in the Introduction section is not clear, please improve this section to strengthen the novelty.
2. The scale bars are missing in the TEM images. Figs. 3-5.
3. I suggest the authors improve the XRD data through normalization of the intensity.
4. Since the nanoparticles have small sizes, how to separate it after use in real wastewater?
5. The conclusion section should be improved; some data are needed.
6. Some data is needed in the Introduction section.
Reviewer 2 Report
The manuscript entitled « Development and upscaling of SiO2@TiO2 core-shell nanoparticles for wastewater treatment purposes » focused on preparation and characterization of core-shell based on SiO2 and TiO2 for photolytic degradation of MB. The obtained results are not original and this work is not novel. Therefore, the authors could supplemented the novelty of this work before possible reconsideration for publication.
1) The “ wastewater treatment” should be deleted from the title and replaced by “ for MB degradation.
2) Results on the photocatalytic degradation should be added in the abstract and conclusion.
3) All figure are in low quality and must be improved
4) More results on the photocatalytic degradation of MB could be added such as the conversion rate.
5) Results on the MB degradation must be compared with other results published elsewhere
Reviewer 3 Report
Authors report on the preparation of SiO2@TiO2 core-shell nanoparticles and elaborate on photocatalytic properties of the materials combined with other characterizations. The work can add to the current body of knowledge; but I have few comments as shown below:
1. Introduction discusses the problems associated with antibacterial resistance, but this topic has nothing to this research. The focus should be on wastewater treatment, dye decomposition and photocatalysis.
2. Despite the long Introduction section, novelty of work is unknown. Please modify the Introduction section to clearly discuss the novelty.
3. In Fig. 3 and other figures, different panels should be labelled and discussed in the text. Scale bars can not be seen and need to be exhibited clearly. Why micrograph of top-left shows a different morphology than that of down-left?
4. In Figure 4, scale bars should be displayed clearly.
5. I suggest to combine Figures 3 and 4, and remove similar micrographs that add nothing more to article.
6. In Figure 5, the scale bars should be clearly displayed. Also there are many micrographs that show same features. Please reduce the number of micrographs. Also I suggest to combine micrographs of same material in one figure.
7. Figure 14, the Time axis should display from 0 min. Some samples show the values close to zero in entire displayed range. Smaller time frames starting from zero are needed to show the results.
8. What are the adsorption performances of various samples?
9. What are the mechanisms involved in photocatalytic degradation of dyes using prepared samples?
10. Please cite alternative nanostructured materials used for degradation methylene blue (MB) including https://doi.org/10.1016/j.jallcom.2021.161201; https://doi.org/10.1016/j.matchemphys.2021.125565; https://doi.org/10.1016/j.inoche.2022.109597; https://doi.org/10.1016/j.jwpe.2023.103903
Generally good; but minor corrections/edit can increase quality of presentation.
Round 2
Reviewer 1 Report
Accept.
Author Response
We thank the reviewer for the suggestions that helped us improve our manuscript.
Reviewer 2 Report
1) The references number of the introduction section could be reduced up to 30-40. Be sure that this work is an original article and not review.
2) I am not agree that this work is on wastewater treatment, but just photodegradation of MB. To suggest wastewater treatment, the material should be evaluated for a concrete/real application of polluted water. However, writing wastewater treatment via the consideration of a simple classic removal of MB prepared in water is so far to elucidate its suitability for wastewater treatment. I suggest that authors can conclude, that prepared material can be used for wastewater treatment AS PROSPECT. If not, the utilization of prepared materials should be employed for real application.
3) The section, 3.6. (Adsorption of probe molecules), authors could add more references to confirm the suggested conclusion about the adsorption of molecules.
4) Table.6 : authors could add the own results obtained in this work
Round 3
Reviewer 2 Report
Now, the revised manuscript can be accepted as is.
Author Response

(The authors gave the same response as above.)
